# Effect of Fermented Concentrate on Ruminal Fermentation, Ruminal and Fecal Microbiome, and Growth Performance of Beef Cattle

**DOI:** 10.3390/ani13233622

**Published:** 2023-11-23

**Authors:** Seul Lee, Chae Hwa Ryu, Youl Chang Back, Sung Dae Lee, Hyeran Kim

**Affiliations:** Animal Nutrition and Physiology Division, National Institute of Animal Science, Rural Development Administration, Wanju 55365, Republic of Korea; chryu0629@korea.kr (C.H.R.);

**Keywords:** fermented concentrate, beef cattle, growth performance, rumen fermentation, microbial diversity

## Abstract

**Simple Summary:**

Fermented feed is utilized to promote the digestion and absorption of nutrients in animal diets while enhancing the host’s immune system and overall health. Fermented feed has been shown to increase the efficiency of nutrient digestion and absorption in livestock, thereby reducing waste production. Additionally, fermentation helps eliminate pathogenic microorganisms that may be present in the feed. The effects of fermented feed, which often contains beneficial microorganisms such as probiotics and yeast, have been extensively studied in various animal species. In ruminant animals, research has predominantly focused on the effects of fermented silage, with limited studies on the feeding effects of fermented concentrate. In this study, we show that fermenting formulated concentrate can have a positive impact on the growth and health of ruminant livestock.

**Abstract:**

The impact of fermented concentrate on the growth and rumen health of beef cattle remains an area of emerging research. This study aimed to assess the influence of a fermented concentrate (TRT) compared to a conventional concentrate (CON) on the growth, rumen fermentation characteristics, and microbiota composition in Korean cattle. Using a crossover design, eight cattle were alternately fed TRT and CON diets, with subsequent analysis of feed components, rumen fermentation parameters, and microbial profiles. TRT and CON diets did not differ significantly in their effect on animal growth metrics. However, the TRT diet was associated with reduced digestibility of rapidly degradable carbohydrates and modified rumen fermentation patterns, as evidenced by an elevated pH and increased acetate-to-propionate ratio (*p* < 0.05). Furthermore, the TRT diet increased the abundance of lactic acid bacteria, *Bacillus*, and yeast and organic acid levels in the rumen (*p* < 0.05). Moreover, Lachnospiraceae and Bacteroidales populations in the rumen and fecal *Akkermansia* abundance increased in the TRT group compared to the CON group. These microbial changes suggest a potential enhancement of the immune system and overall health of the host. Further research on the long-term implications of incorporating fermented concentrate into cattle diets is warranted.

## 1. Introduction

Use of fermented feed in livestock aims to enhance digestion and absorption by processing low-quality feed and supplying it to animals, ultimately promoting the well-being of the host. When fermented feed is used, costs are significantly reduced by maximizing feed efficiency and reducing the amount of feces produced due to high nutrient digestion and absorption [1]. Fermentation is the process by which microorganisms convert starch and sugar into fermentation products, such as lactic acid, organic acids, and alcohols. At the beginning of fermentation, acidity is high because the levels of lactic acid bacteria, yeast, and lactic acid are generally low. During the stabilization period, levels of lactic acid bacteria (LAB) such as *Lactobacillus* increase and acidity decreases [2]. Feed fermentation is also an effective method used to kill residual pathogenic microorganisms that may remain in feed [3]. In addition, fermented feed can lower gastric acidity, promote pancreatic juice secretion, and improve digestion and absorption of nutrients [4]. Silage and fermented total mixed ration (TMR), which are types of fermented feed, have been used in ruminant breeding management programs [5]. When producing fermented feed, microorganisms such as LAB and yeast are used that typically improve feed quality. The addition of microorganisms to fermented feeds alters the fermentation properties of the rumen [6] and improves the crude protein (CP) and fiber degradation rate [7]; moreover, it improves digestion [8] and increases palatability and intake [5]. In particular, because fungi produce various protein- and carbohydrate-decomposing enzymes, such as amylase, xylanase, and protease, their cultures are widely used in fermented feeds [9]. Enzymes produced by fungi improve feed value by increasing substrate availability [10]. In addition, they supplement the action of endogenous enzymes in animals to promote the absorption of nutrients into the body [11]. However, because aerobic spoilage is usually initiated by yeasts, care must be taken to prevent spoilage when yeast is added [12].

Although previous studies have acknowledged the nutritional benefits of fermented feeds [1], there has been a predominant research focus on monogastric animals, including pigs and poultry, often within the realms of silage or TMR. In contrast, investigations into the effects of fermentation on feed, especially concentrate formulations for ruminant species, are comparatively underrepresented in the literature. This research aims to contribute to this underexplored area by examining the quality of fermented concentrates and assessing their influence on growth performance, ruminal fermentation patterns, and microbial diversity in beef cattle.

## 2. Materials and Methods

### 2.1. Experimental Design and Animal Management

All of the animal experimental procedures were reviewed and approved by the Institutional Animal Care and Use Committee of the National Institute of Animal Science (No. 2021-501).

Eight Hanwoo steers (25 months old, 603 ± 89.4 kg) participated in a 110-d feeding trial at a beef cattle farm of the National Institute of Animal Science, Rural Devel-opment Administration. Two cattle with similar body weights (BWs) were grouped and housed in a pen. Each pen (5 m × 5 m) was equipped with a feed bin. The animals were fed twice daily at 09:00 and 16:00 with each experimental diet, the amount of which was adjusted to achieve 10% refusal based on previous intake. The forage utilized a blend of hay produced by mixing Tall Fescue and Kentucky Bluegrass in a 7:3 ratio. Forage contained 90.1% dry matter (DM), 8.5% CP, 6.0% crude fat, 69.1% neutral detergent fiber analyzed using heat-stable amylase and expressed inclusive of residual ash (aNDF), 45.0% acid detergent fiber (ADF), 15.7% non-fiber carbohydrates (NFCs), and 8.1% ash. Drinking water was freely accessible.

The cattle were randomly assigned to one of the two treatment sequences in a switchback design with three 30-d periods and a 10-d washout period. Each period consisted of 28 d for diet adaptation and data collection (feed intake) and 2 d for sample collection (rumen, feces) and data collection (BW). The experimental treatments comprised conventional concentrate (CON) and fermented concentrate (TRT). Both the CON and TRT groups were given the same forage, with the difference lying in the concentrate feed. The CON group was administered concentrate in its powdered form, whereas the TRT group was given the same concentrate that had been fermented for three days in a fermentation concentrate manufacturing machine made of stainless steel (EO-1500, EO tech, Gwangju, Republic of Korea) set at 35 °C, with water added to achieve a moisture content of 40%. In our research, we used the EO-1500 fermentation machine, which has a 1500-L capacity and is constructed from durable stainless steel (SUS304). Its design includes a triple-layer structure for efficient heat retention and a double ribbon mixer for effective content circulation. The machine also features a dual temperature control system and a unique convection-based drying method, ensuring optimal fermentation conditions. During production of the fermented concentrate, we utilized a microbial complex product (*Bacillus subtilis* 1.0 × 10^6^ cfu/g, *Enterococcus faecium* 1.0 × 10^6^ cfu/g, *Saccharomyces cerevisiae* 1.0 × 10^6^ cfu/g; Bio 5050, Nonghyup Feed Gunsan Bio, Gunsan, Republic of Korea). Ultimately, a probiotic was added at a level of 0.25% of the final fermented concentrate weight. All cattle were placed completely randomly. Experimental animals received both diets throughout the three experimental periods in the order TRT-CON-TRT or CON-TRT-CON (switchback design) [13,14]. Daily DM intake was measured throughout the feeding trial. Individual daily feed intake was recorded by measuring the feed offered and refusals. Before providing the diet to the animals, we measured the moisture content and feed weight of the raw feed. Each morning, we collected the uneaten refusal diet and measured the moisture content and weight of the raw refusal diet. Moisture measurements of both the diet given to the animals and the remaining diet were taken after sub-sampling following thorough mixing, with three replicates. BW was measured at the start and end of the period before morning feeding.

### 2.2. Formula and Chemical Composition of Experimental Diets

The formulas and chemical compositions of the experimental diets are summarized in Table 1 and Table 2. The vitamin complex used in this study consisted of 2,650,000 IU vitamin A, 530,000 IU vitamin D3, 1050 IU vitamin E, 10 g nicotinic acid, 4.4 g manganese sulfate, 4.4 g zinc sulfate, 13.2 g ferrous sulfate and ferric oxide, 2.2 g cupric sulfate, 0.44 g calcium iodate, and 0.44 g cobaltous carbonate per kg. The fermented concentrate was prepared by adding probiotics and moisture to initiate fermentation. 

All feed samples used in the experiment were dried at 60 °C for 48 h and ground in a cyclone mill (Foss, Hillerød, Denmark) fitted with a 1-mm screen. The DM, ADF, ash, and ether extract (EE) were analyzed using the procedure described by Horwitz [15]. NDF was analyzed using heat-stable amylase and was expressed inclusive of residual ash [16]. Total nitrogen content was measured via the Dumas combustion method [17] using an elemental combustor (Vario Max Cube, Elementar Gmbh, Frankfurt, Germany), and CP content was calculated as 6.25 times the nitrogen content. The acid detergent-insoluble CP (ADICP) and neutral detergent-insoluble CP (NDICP) levels in each sample were determined as described by Licitra et al. [18]. NFC level was calculated based on the guidelines provided by the National Research Council [19].
NFC (%DM) = 100 − ash − EE − CP − (aNDF − NDICP)(1)

### 2.3. Calculation of Degradable Carbohydrate and Protein Fractions

The formulae for calculating carbohydrate (CHO) and protein digestibility in the rumen are as follows [20]:CA (%CHO) = CHO − CB1 − CB3 − CC(2)
CB1 (%CHO) = starch(3)
CB2 (%CHO) = NFC − CA − CB1(4)
CB3 (%CHO) = aNDF − NDICP × CP/1000 − CC(5)
CC (%CHO) = aNDF × Lignin × 2.4/1000(6)

CA: carbohydrate A fraction, instantaneously degradable carbohydrates.CB1: carbohydrate B1 fraction, starch.CB2: carbohydrate B2 fraction, intermediately degradable carbohydrates.CB3: carbohydrate B3 fraction, slowly degradable carbohydrates.CC: carbohydrate C fraction, unavailable cell wall.

PA (%CP) = non-protein nitrogen (NPN) (%SOLP) × 0.01 × SOLP (%CP)(7)

PB1 (%CP) = SOLP (%CP) − PA(8)

PB2 (%CP) = 100 − PA − PB1 − PB3 − PC(9)

PB3 (%CP) = NDICP (%CP) − ADICP (%CP)(10)

PC (%CP) = ADICP (%CP)(11)

PA: protein A fraction, instantaneously solubilized protein.SOLP: soluble CP.PB1: protein B1 fraction, rapidly degradable protein.PB2: protein B2 fraction, intermediately degradable protein.PB3: protein B3 fraction, slowly degradable protein.PC: protein C fraction, completely undegradable protein.

### 2.4. Fermentation Feed Quality Analysis

The feed samples (10 g) and distilled water (90 mL) were mixed and homogenized for 2 min in a stomacher (Wisemix^®^, Daihan, Republic of Korea). After centrifugation (2300× *g*; 4 °C; 20 min), the supernatant was used for pH and ammoniacal nitrogen content analyses. The pH of the samples was analyzed using a pH meter (Seven Easy; Mettler Toledo^®^, Columbus, OH, USA). Ammoniacal nitrogen content was determined according to the method described by Chaney and Marbach [21]. The phenol color reagent (1 mL) and alkali hypochlorite reagent (1 mL) were mixed with 20 µL of the supernatant. After reacting at 37 °C for 15 min, the absorbance was measured at 630 nm using a spectrophotometer (Optizen UV2120, Mecasis, Republic of Korea). Each fermentation sample (10 g) was mixed with 90 mL peptone water (Difco, Detroit, MI, USA). After homogenizing for 2 min using a stomacher (Wisemix^®^, Daihan, Republic of Korea), the homogenate was subjected to organic acid analysis via HPLC (Varian, Palo Alto, CA, USA) utilizing a C18 column. Sample preparation was performed by dissolving 0.1 g of the sample in 20 mL of 0.4% hydrochloric acid, followed by ultrasonication. HPLC conditions included isocratic pumping, a mobile phase of 520 mM H_2_SO_4_, UV detection at 210 nm, a flow rate of 1.0 mL/min, and a 20 µL injection volume. Quantification was based on the formula: Organic acid (%) = (sample peak area/standard peak area) × (concentration of standard solution (g/50 mL)/sample weight (g)) × 100. Viable cell counts were determined as described by Miller and Wolin [22]. LAB and *Bacillus* were cultured for 48 h in an incubator at 37 ± 1 °C using De Man, Rogosa, and Sharpe (Difco, Detroit, MI, USA) and Luria–Bertani (Difco, Detroit, MI, USA) media, respectively. The yeast cells were cultured for 48 h in malt chloramphenicol. Thereafter, the number of viable cells was determined by counting the number of colonies formed on each plate.

### 2.5. Sample Collection and Analysis

Fresh fecal samples were obtained from the cattle on day 29 of each experimental period. The samples were immediately stored at −80 °C until metagenomic DNA extraction.

On day 30 of each experimental period, representative rumen fluid samples were collected via a stomach tube approximately 3 h after feeding [23,24]. Between samples, the stomach tube was thoroughly washed with warm water to prevent cross-contamination from the previous animal [25,26]. The first 200 mL of ruminal fluid was discarded to reduce contamination by saliva. Immediately after collection, the pH of the sampled inoculum was measured using a pH meter (Pinnacle pH meter M540; Corning, NY, USA). The ruminal fluid was then sealed in a tube and frozen in liquid nitrogen. The samples were stored at −80 °C until the analysis of volatile fatty acid (VFA) and ammonia nitrogen (NH_3_-N) levels and metagenomic DNA extraction. The VFA and NH_3_-N concentrations were determined as described by Erwin et al. [27] and Chaney and Marbach [21] with minor modifications.

The ruminal fluid was centrifuged at 6000× *g* for 15 min at 20 °C. The supernatant was used for VFA analysis. For VFA analysis, a 25% metaphosphoric acid solution was added to the ruminal fluid at 10% of the total volume. The supernatant was injected into a gas chromatograph (TRACE 1610, Thermo Fisher Scientific, Waltham, MA, USA) equipped with a flame ionization detector and capillary column (Nukol™, fused silica capillary column 15 m × 0.53 mm × 0.5 µm; Supelco Inc., Bellefonte, PA, USA). The oven, injector, and detector temperatures were 110 °C, 250 °C, and 250 °C, respectively. A standard curve was generated using a VFA standard solution (catalog number. 46975-U; Sigma-Aldrich, St. Louis, MO, USA). The mixtures of inoculum and 25% metaphosphoric acid were centrifuged at 14,000× *g* for 5 min at 4 °C for NH_3_-N analysis. After centrifugation, 20 µL of each supernatant was mixed with 1 mL of phenol color reagent (50 g/L of phenol plus 0.25 g/L of nitroferricyanide) and 1 mL of alkali hypochlorite reagent (25 g/L of sodium hydroxide and 16.8 mL/L of 4–6% sodium hypochlorite). The mixture was incubated in a water bath for color development at 37 °C for 15 min; thereafter, 8 mL of distilled water was added, and the NH_3_-N concentration was determined by measuring the absorbance at 630 nm using a UV spectrophotometer (Bio-Rad, US/benchmark plus, Tokyo, Japan). All analyses were repeated three times, and the mean values are presented.

### 2.6. Microbial Diversity Analysis

Metagenomic DNA was extracted from the collected fecal and ruminal fluid samples. For metagenome community analysis, the V3–V4 region of the 16S rRNA was used as a phylogenetic marker. The amplicons were generated using the 337 F and 805R 16S V3–V4 rRNA universal primers (GACTCCTACGGGAGGCWGCAG and GACTACCAGGGTATCTAATC). Sequencing libraries were constructed using Herculase II Fusion DNA Polymerase Nextera XT Index Kits by Agilent Technologies (Santa Clara, CA, USA). All library construction processes were conducted following the 16S Metagenomic Sequencing Library Preparation guidelines by Illumina. Before conducting analysis, initial quality control was performed using Trimmomatic [28] to remove low-quality base calls and sequencing artifacts. The parameters used were ILLUMINACLIP:TruSeq3-PE.fa:2:30:10:2:True LEADING:5 TRAILING:20 MINLEN:250. Microbial community analysis was performed using the QIIME2 [29] pipeline with a DADA2 [30] and SILVA [31] pre-trained Naive Bayes classifier based on the SILVA full-length 16S rRNA database. A Kruskal–Wallis test and linear discriminant analysis effect size [32] were used to compare the effect of the fermented concentrate by treatment on the rumen and intestine.

### 2.7. Statistical Analysis

Data for the experimental diets, growth performance, and rumen fermentation parameters satisfied the conditions of normality and homoscedasticity for each group and were analyzed using descriptive statistics and the *t*-test in SPSS (Version 26, IBM, Armonk, NY, USA). Statistical significance was set at *p* < 0.05.

## 3. Results and Discussion

TRT diet with water added for fermentation had a significantly lower DM content than that in the CON diet (*p* < 0.05; Table 1). However, there were no significant differences between the treatments in terms of the composition of CP, crude fat, aNDF, and others on a DM basis. The digestibility of carbohydrates and proteins was calculated based on the chemical composition of the experimental diets (Table 3). The proportion of instantaneously degradable carbohydrates (CA), which are immediately digested in the rumen, was significantly lower in the TRT diet than that in the CON diet (*p* < 0.05). This difference may be attributed to the fact that microorganisms had already decomposed some carbohydrates during the concentration process in TRT. There was no significant difference between the treatment groups in terms of the degradable carbohydrate fraction, which is classified according to the carbohydrate decomposition rate, and the unavailable cell wall fraction in the rumen. Furthermore, there was no significant difference in protein digestibility among the instantaneously solubilized protein, degradable protein, and completely undegradable protein fractions, regardless of the rumen digestion rate.

In general, feed fermentation can be determined by microbial population and organic acids [33]. *Enterococcus faecium*, a homofermentative lactic acid bacterium, is used as an additive that can increase lactic acid production in silage [34]. Owing to the presence of microbial additives, the TRT diet had significantly higher abundances of LAB, *Bacillus*, and yeast than the CON diet (*p* < 0.05, Table 4). The pH was significantly lower in the TRT group than that in the CON diet (*p* < 0.05). This finding is consistent with results of previous studies reporting that the addition of the homofermentative LAB lowers the pH during silage fermentation [35,36]. pH and acetic acid levels are used as indicators to confirm fermentation stability in fermented feeds [37]. The acetic acid level was significantly higher in the TRT diet than that in the CON diet (*p* < 0.05). In addition, the TRT diet exhibited significantly higher levels of organic acids, such as lactic acid, propionic acid, and butyric acid, than the CON diet did, indicating a stable fermentation in the former. Ammonia nitrogen production was also significantly higher in the TRT diet than that in the CON diet (*p* < 0.05). During the fermentation of the concentrate, proteins and other nitrogen-containing compounds are broken down by microbes, resulting in the conversion to NH_3_-N. The concentration of NH_3_-N during this process can serve as a critical indicator of the fermentation quality. Maintaining appropriate levels of ammonia nitrogen in fermented feeds is important because it affects protein metabolism and energy balance within the rumen. By ensuring that these levels are kept within optimal ranges, the productivity and health of ruminants can be optimized [38].

After the experimental trial period, there was no significant difference in the final BW, daily weight gain, feed intake, or feed conversion ratio between the treatments (Table 5). Although the mean daily weight gain was higher in the TRT group (0.96) than in the CON group (0.67), the difference was not statistically significant. The large standard error associated with the weight gain data (0.105) indicates a considerable variability within the groups, thereby precluding a definitive conclusion regarding the effect of the treatment on weight gain. Fermentation did not affect the chemical composition of the feed or the amount of feed consumed; thus, it had no effect on nutrient intake.

Ruminal pH was significantly higher in the TRT group than in the CON group (*p* < 0.05; Table 6). There were no significant differences in the NH_3_-N and total VFA levels between treatments; however, acetic acid, propionic acid, and butyric acid production significantly differed between the groups (*p* < 0.05). The acetic acid and lactic acid levels were higher and propionic acid levels were lower in the TRT group than those in the CON group (*p* < 0.05). The acetate to propionate ratio (AP) ratio was significantly higher in the TRT group than that in the CON group (*p* < 0.05). In accordance with our research findings, a study that fed fermented soybean meal to Holstein cows also reported an increase in the acetate percentage, pH, and AP ratio in the rumen fluid [39]. Conversely, in another study, Holsteins fed fermented soybean meal showed no changes in the ruminal pH and an increase in the AP ratio [40]. In light of our research results, the higher rumen pH and elevated AP ratio observed in the TRT group may be correlated, aligning with the findings of Amin [39] and Russell [41]. In the context of beef cattle experiencing a rapid increase in BW and marbling score just before slaughter due to excessive concentrate feeding, a sharp decline in rumen pH can lead to an increased risk of acidosis. In our study, despite a relatively high proportion of concentrate intake within the total diet, accounting for 90.6% in the CON group and 86.5% in the TRT group, the fermented concentrate resulted in an elevation of the rumen pH. Therefore, using fermented concentrate during periods of excessive concentrate feeding may raise the rumen pH, potentially preventing acidosis.

In response to the administration of fermented concentrate, there was a statistically significant increase in the alpha diversity of rumen microbiota, as indicated by the higher Shannon entropy values, in the TRT group compared to the CON group (*p* < 0.05, Figure 1). This finding suggests that the intake of fermented concentrate results in increased microbial diversity within the rumen. Conversely, when comparing the alpha diversity of the fecal microbiome across different treatments, no statistically significant differences were observed. In terms of beta diversity analysis, rumen microbiota displayed relatively distinct clustering patterns among the groups, highlighting clear separations. In contrast, fecal microbiota did not exhibit discernible clustering patterns among the groups. In summary, the use of fermented concentrate exerts a more pronounced impact on rumen microbiota than that on the post-rumen digestive intestinal. Fermented feed primarily affects the rumen microbiome in ruminants, enhancing the microbial activity, which is crucial for digestion [42,43]. This modulation can improve feed efficiency and nutrient uptake. The fecal microbiome is less directly impacted by fermented feed, reflecting post-digestive processes. However, diet can influence both the rumen and fecal microbiome, with more pronounced changes in the former.

The abundance of rumen microbial clusters belonging to the family Lachnospiraceae increased in cattle administered the fermented concentrate (Figure 2). Lachnospiraceae is commonly found in cattle rumen and plays a role in fermenting plant polysaccharides to produce short-chain fatty acids and alcohol. It is also associated with disease resistance because it is involved in the production of butyric acid, a critical substance for microbe–host epithelial cell growth, which can mitigate intestinal inflammation and maintain the intestinal barrier [44]. However, there are conflicting findings regarding the impact of Lachnospiraceae on livestock productivity. Previous analyses of rumen microbes in cattle with low nitrogen efficiency showed high levels of Lachnospiraceae [44,45]. In contrast, the abundance of the genus *Moryella* belonging to Lachnospiraceae, which has been linked to improved feed efficiency in livestock, was found to be increased in the TRT group. *Moryella* was the dominant genus in the rumen of calves with low residual feed intake. Additionally, inoculation of the rumen with *Moryella* led to a significant increase in its abundance, potentially enhancing propionate production capacity and improving feed efficiency [46,47]. In this study, the altered abundance of previously reported taxa did not directly impact weight gain or feed efficiency.

In the TRT group, an increase in the abundance of ruminal microorganisms related to the breakdown of feed nutritional components was observed. Genera belonging to the order Bacteroidales showed a significant increase in the TRT group compared to the CON group. Bacteroidales are involved in the decomposition and fermentation of plant fibers, as well as the breakdown of complex polysaccharides and other complex fibers into simple organic compounds [48]. Furthermore, the abundance of Bradymonadales and *Desulfovibrio*, belonging to the phylum Termodesulfobacteriota, which is known for its involvement in sulfate reduction and digestion of organic matter, increased in the TRT group compared to the CON group [49]. Bradymonadales, although relatively less known, preferentially prey on Bacteroidetes and Proteobacteria [50].

Analysis of fecal microbiota revealed an increase in the abundance of *Akkermansia* in the TRT group compared to the CON group (Figure 2). *Akkermansia*, a slender-shaped bacterium, is commonly found in bovine feces and is particularly abundant in individuals consuming a high forage diet [51,52,53]. Furthermore, *Akkermansia* maintains a healthy gut microbiota and enhances immune function, potentially aiding in disease prevention. Moreover, it plays a positive role in preventing metabolic disorders as a member of the gut microbiota and is often highly abundant in healthy humans [54,55]. However, additional research is needed to understand the potential immunomodulatory and disease-preventing effects of fecal *Akkermansia* in cattle, particularly regarding its impact on host health.

Our study acknowledges the limitation of single time point collection in reflecting the rumen’s dynamic nature, and suggests the potential for more comprehensive insights through multi-time point sampling in future research. We recognize the possible influence of using a stomach tube on rumen fluid characteristics, and have detailed our mitigating measures in the Methods section, acknowledging the need for careful consideration of these factors in interpreting our results.

## 4. Conclusions

This study investigated the effect of fermented concentrate feed on its nutritional composition and on beef cattle. Fermentation led to a decrease in the CA fraction of the concentrate but no significant changes in the levels of other components. Microbial abundance increased during fermentation. Stable fermentation resulted in higher organic acid levels and a lower pH, potentially improving rumen digestibility. Fermented concentrate treatment did not significantly affect weight gain, feed conversion ratio, or final BW, but did improve the rumen fermentation characteristics, including pH and acetic acid levels, indicating stable rumen fermentation. Our findings suggest that fermented concentrate primarily affects the rumen microbial community rather than the gastrointestinal tract. Fermented concentrate treatment increases the abundance of microorganisms enhancing the feed digestibility, immunity, and health of the host. While fermented concentrate does not impact cattle growth and digestibility in the short term, it may have long-term effects on cattle growth and farm economics. Future research with a greater number of sampling days is needed to explore these potential long-term benefits.

## Figures and Tables

**Figure 1 animals-13-03622-f001:**
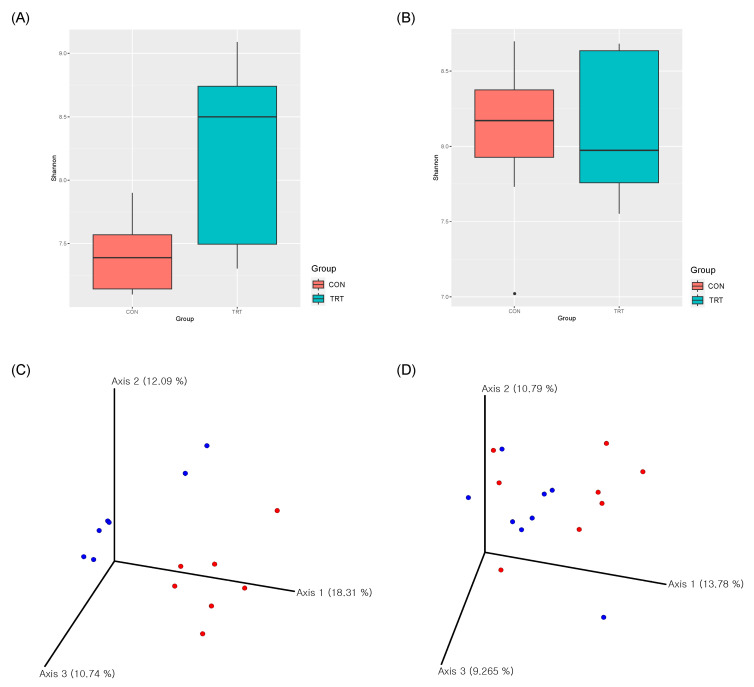
Ruminal and fecal bacterial community diversities. Alpha diversity in the ruminal bacterial community (**A**) and fecal bacterial community (**B**). Principal coordinate analysis plots for the ruminal bacterial community (**C**) and fecal bacterial community (**D**); each dot in the plots represents a cluster. Samples were collected from Hanwoo steers fed conventional concentrate (CON; *n* = 8, represented by red dots) and fermented concentrate (TRT; *n* = 8, represented by blue dots).

**Figure 2 animals-13-03622-f002:**
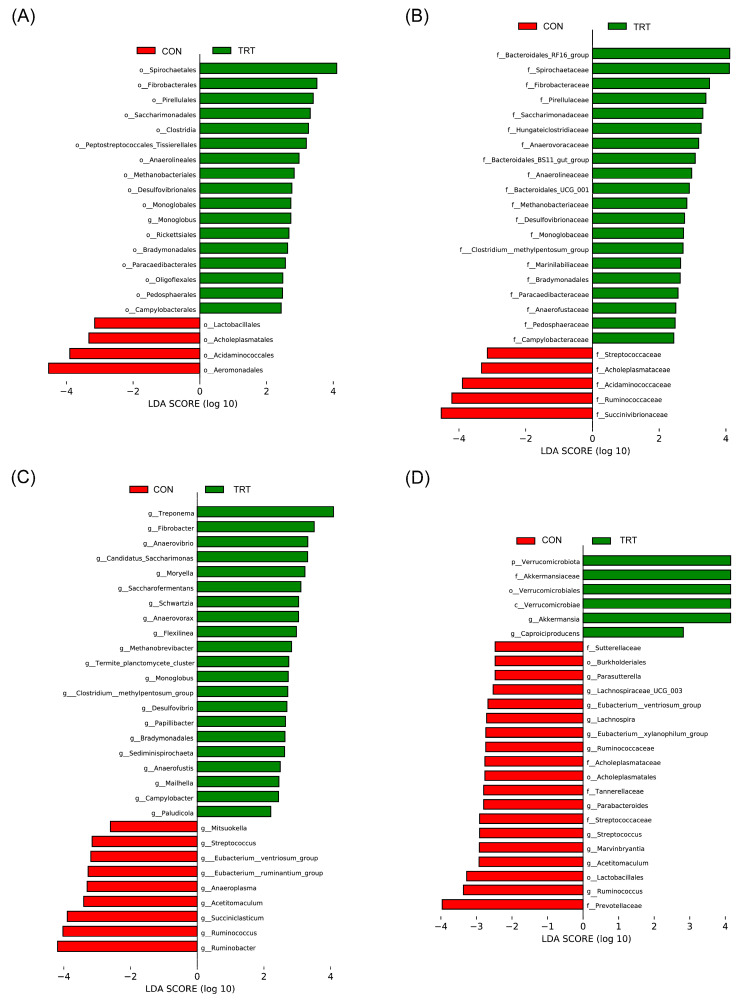
Linear discriminant analysis of the (**A**–**C**) rumen and the (**D**) feces. Samples were collected from Hanwoo steers fed conventional concentrate (CON; *n* = 8) and fermented concentrate (TRT; *n* = 8).

**Table 1 animals-13-03622-t001:** Concentrate feed ingredients.

Items	Ingredients (% as-Fed Basis)
Corn flakes	35.7
Wheat bran	19.6
Corn gluten feed	12.5
Wheat	8.8
Soybean meal	7.8
Palm meal	6.3
Soybean hull	3.7
Lupin flake	3.7
Limestone	0.9
Salt	0.4
Sodium bicarbonate	0.4
Vitamin complex	0.1
Total	100.0

**Table 2 animals-13-03622-t002:** Chemical composition of experimental diets.

Items	CON Diet	TRT Diet	SEM	*p*-Value
Dry matter, %	90.39	61.59	3.018	<0.05
⋯⋯⋯⋯⋯⋯⋯⋯⋯⋯⋯⋯⋯⋯⋯⋯⋯⋯% dry matter⋯⋯⋯⋯⋯⋯⋯⋯⋯⋯⋯⋯⋯⋯⋯⋯⋯⋯				
CP	18.33	18.93	0.301	0.375
Soluble CP	7.60	8.40	0.342	0.286
Neutral detergent-insoluble CP	1.52	1.52	0.088	1.000
Acid detergent-insoluble CP	0.83	0.92	0.033	0.185
Ether extract	2.94	2.74	0.139	0.520
Neutral detergent fiber ^(1)^	28.13	26.50	1.316	0.595
Acid detergent fiber	12.50	10.80	0.753	0.307
Non-fiber carbohydrate	46.90	47.50	1.416	0.858
Ash	5.24	5.84	0.220	0.203

CON diet, conventional concentrate without microbial additive; CP, crude protein; TRT diet, fermented concentrate with microbial additive; SEM, standard error of the mean. ^(1)^ NDF, neutral detergent fiber analyzed using heat-stable amylase and expressed inclusive of residual ash.

**Table 3 animals-13-03622-t003:** Degradable carbohydrate and protein fractions of experimental diets (dry matter basis).

Items	CON Diet	TRT Diet	SEM	*p*-Value
Carbohydrate fractions ^(1)^, %CHO				
CA	4.96	1.24	0.932	<0.05
CB1	43.64	46.26	2.733	0.683
CB2	15.17	18.03	1.002	0.173
CB3	28.59	27.65	1.443	0.783
CC	7.68	6.79	0.534	0.468
Protein fractions ^(2)^, %CP				
PA + PB1	41.46	44.34	1.466	0.383
PB2	50.28	47.60	1.336	0.374
PB3	3.73	3.18	0.530	0.662
PC	4.54	4.88	0.191	0.443

CON diet, conventional concentrate without microbial additive; TRT diet, fermented concentrate with microbial additive; SEM, standard error of mean. ^(1)^ CHO, carbohydrate; CA, instantaneously degradable carbohydrates; CB, degradable carbohydrates; CB1, starch; CB2, intermediately degradable carbohydrates; CB3, slowly degradable carbohydrates; CC, unavailable cell wall. ^(2)^ CP, crude protein; PA, instantaneously solubilized protein; PB, degradable protein; PB1, rapidly degradable protein; PB2, intermediately degradable protein; PB3, slowly degradable protein; PC, completely undegradable protein.

**Table 4 animals-13-03622-t004:** Fermentation and microbial profile of the experimental diets (as-fed basis).

Items	CON Diet	TRT Diet	SEM	*p*-Value
Microbial profile, log10 cfu/g				
Lactic acid bacteria	4.26	8.46	0.579	<0.05
*Bacillus*	4.37	5.14	0.110	<0.05
Yeast	4.00	6.41	0.439	<0.05
Fermentation profile				
pH	5.69	4.40	0.150	<0.05
Lactic acid, %	0.12	0.24	0.028	<0.05
Acetic acid, mM	0.34	18.31	2.897	<0.05
Propionic acid, mM	0.00	0.66	0.109	<0.05
Butyric acid, mM	0.00	0.26	0.067	<0.05
Ammonia nitrogen, mg/dL	0.63	2.61	0.259	<0.05

CON diet, conventional concentrate without microbial additive; TRT diet, fermented concentrate with microbial additive; Forage, mixed hay with Tall Fescue and Kentucky Bluegrass in a 7:3 ratio; SEM, standard error of mean.

**Table 5 animals-13-03622-t005:** Effects of fermented concentrate on growth performance of Hanwoo steers (dry matter basis).

Items	CON	TRT	SEM	*p*-Value
Growth performance				
Initial body weight, kg	655.66	666.09	33.506	0.888
Final body weight, kg	687.66	690.41	34.343	0.971
Average daily gain, kg/d	0.96	0.67	0.105	0.169
Feed intake, kg/d	11.61	12.18	0.479	0.568
Concentrate intake, kg/d	9.98	10.53	0.425	0.558
Forage intake, kg/d	1.63	1.65	0.124	0.929
Feed conversion ratio	16.13	22.99	2.311	0.175
Nutrient intake, kg/d				
Crude protein	1.97	2.21	0.076	0.114
Ether extract	0.31	0.31	0.010	0.785
Neutral detergent fiber	3.93	4.02	0.118	0.719
Non-fiber carbohydrate	4.94	5.46	0.186	0.173
Gross energy	51.39	55.82	1.784	0.233

CON, conventional concentrate without microbial additive; TRT, fermented concentrate with microbial additive; SEM, standard error of mean.

**Table 6 animals-13-03622-t006:** Effects of fermented concentrate on rumen fermentation characteristics of Hanwoo steers.

Items	CON	TRT	SEM	*p*-Value
pH	6.27	6.60	0.056	<0.05
Ammonia nitrogen, mg/dL	11.13	16.33	3.019	0.348
Total volatile fatty acid, mM	113.82	103.83	6.347	0.566
Acetate, %	60.67	64.00	1.074	<0.05
Propionate, %	25.56	19.18	1.677	<0.05
Butyrate, %	10.35	13.03	0.533	<0.05
Valerate, %	3.42	3.79	0.173	0.227
Acetate-to-propionate ratio	2.44	3.39	0.242	<0.05

CON, conventional concentrate without microbial additive; TRT, fermented concentrate with microbial additive; SEM, standard error of the mean.

## Data Availability

The data presented in this study are available on request from the first author. The data are not publicly available due to restrictions by the research group.

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
