# Peer review of "Effect of Fermented Concentrate on Ruminal Fermentation, Ruminal and Fecal Microbiome, and Growth Performance of Beef Cattle"

_animals, 2023, doi:10.3390/ani13233622_

Round 1

Reviewer 1 Report

Comments and Suggestions for Authors

This manuscript aimed to investigate effects of fermented concentrate on growth performance, rumen fermentation, and microbial diversity of beef cattle. The results may help to better understand the fermented feed impact on the growth and health of ruminants. However, the novelty of this study is not important as we expected. In addition, the discussion is not appropriate or not closely related to the results. More comments as follows:

L76-78 Table 1 is the composition of concentration diet. Where is ingredients and chemical composition of forage diet?

 L84-85 “Both the CON and TRT groups were given the same forage”. Is the forage refers to L76 forage ? What is the ratio of forage and concentrate/ fermented feed?

L87 Why the authors choose 3 days at 35°C for concentrate fermentation?  Are these parameters the best fermented conditions for concentrate ?

L95 How to measure the feed intake? Please describe the details.

L90-91 and L108-109 What is the concentration of the probiotics mixture? 1.0 × 106 cfu/g or 2.0 × 106 CFU/g? Please double check it.

L140-157 The method describes the analysis method for the fermentation quality of fermented feed (TRT), while the CON group is unfermented feed and its fermentation parameters are shown in Table 3. How to determined the fermentation characteristics of CON?

L150 The method for analyzing organic acids should be written in more detail.

L152,155,156 Is the word 'cell' accurate?

Line 159 only one day for fecal sampling in each experiment period, the time point is too less.

Line 162 Most rumen contents are rumen fluid samples and less solid contents, it can not represent the real microbial conditions in the rumen.

L233 Is this sentence correct? Please check.

L235-236 Fermentation is the process by microorganism’s converts starch and sugar into organic acids including lactic acid which reduces the pH. The LAB inoculant used at ensiling ensures vigorous fermentation and positively affects silage stabilization, which results in increased accumulation of organic acids. In the current study, I do feel that it is oversimplified, and that a broader discussion of the results is more appropriate.

L254-256 From Table 4, although there were no significant differences in daily weight gain, feed intake, or the feed conversion ratio between the treatments, animals fed TRT have higher Feed intake, Concentrate intake, Forage intake and Feed conversion ratio than those fed con diet. Why does animal daily weight gain decrease? Please explain.

L337-350 Conclusions is excessively long and are only weakly supported by the presented evidence. Please reword.

Reviewer 2 Report

Comments and Suggestions for Authors

This manuscript describes an experiment to develop a new feed production system. There are no major concerns with this study.

 L46 Re-check: lower pH should increase acidity.

L72 recheck, RCBD did not respond to the Switchback design in Lines 80, 94. Which experimental design?

L208 adds information on an experimental statistical analysis model see Lucas (1960)

L243 add discussion trt effect on NH3-N

L254 Table 4 adds a discussion on the limitation of this study eg. stage of growth, short time (days) of the experiment, and experimental design to assess growth performance.

L254 Table 4 specifies which forage type-species.

L343 adds the limitation of this study eg. stage of growth, days of the experiment, experimental design

Reviewer 3 Report

Comments and Suggestions for Authors

Please see comments in the attachment

Comments on the Quality of English Language

Minor improvements

Reviewer 4 Report

Comments and Suggestions for Authors

The purpose of this research was to evaluate the quality of a fermented concentrate (Probiotics), as well as to investigate its effects on growth performance, rumen fermentation and microbial diversity when administered to beef cattle.

However, a more detailed description of the methodology used is recommended, with the following improvements:

-In the "Material and Methods" section, in point 2.3 (line 125), the equations used to determine the degradability of carbohydrates and protein fractions should be accompanied by bibliographical references justifying their choice and validity.

-In point 2.7 (line 206), when describing the statistical test used, it is essential to mention whether the data met the assumptions of normality and homogeneity, as well as explaining the reasons behind the choice of statistical test in relation to the nature of the data.

In this way, these modifications will help to clarify the proposal, methodology and statistical robustness of the study, thus improving the quality and general comprehension of the manuscript.

The section dedicated to results and discussion (line 210) is complex enough to compromise the reader's understanding. It is therefore suggested that a more structured approach be adopted, with the results and discussion sections clearly distinguished and separated. This refinement in the structure of the manuscript will contribute to a more effective and elucidating presentation of the data and subsequent analysis.

Line 305 - the bibliographical references should be numbered, which is not the case with Alves et al. 2021 and Meehan et al. 2014.

Round 2

Reviewer 1 Report

Comments and Suggestions for Authors

The authors should clarify the limitation and potential influence of single time point and take rumen fluid by stomach tube in discussion. 

Author Response

Thank you for your valuable feedback regarding our methodology, specifically the use of a single time point and the collection of rumen fluid via a stomach tube. We acknowledge these concerns and appreciate the opportunity to discuss their implications and our mitigation strategies. We revised discussion part as you suggested. This response acknowledges the limitations, and suggests potential areas for future research, demonstrating your understanding and addressing the reviewer's concerns effectively.

Reviewer 3 Report

Comments and Suggestions for Authors

Your responses to the queries raised are sufficient 

Comments on the Quality of English Language

Minor corrections

Author Response

We would like to express our gratitude for your valuable review of our manuscript. Following your comments, we have made English corrections and are submitting the revised document along with the certificate.

Reviewer 4 Report

Comments and Suggestions for Authors

The authors have made all the suggested changes

Author Response

We would like to express our gratitude for your valuable review of our manuscript.
